# Inclusive Crisis Communication in a Pandemic Context: A Rapid Review

**DOI:** 10.3390/ijerph21091216

**Published:** 2024-09-16

**Authors:** Karin Hannes, Pieter Thyssen, Theresa Bengough, Shoba Dawson, Kristel Paque, Sarah Talboom, Krizia Tuand, Thomas Vandendriessche, Wessel van de Veerdonk, Daniëlle Wopereis, Anne-Mieke Vandamme

**Affiliations:** 1Research Group SoMeTHin’K, Centre for Sociological Research, Faculty of Social Sciences, KU Leuven, 3000 Leuven, Belgium; pieter.thyssen@uclouvain.be (P.T.); danielle.wopereis@viadonbosco.org (D.W.); 2Institute for the Future, KU Leuven, 3000 Leuven, Belgium; annemie.vandamme@kuleuven.be; 3Department of Microbiology, Immunology and Transplantation, Rega Institute for Medical Research, Clinical and Epidemiological Virology, KU Leuven, 3000 Leuven, Belgium; 4Austrian National Public Health Institute, 1010 Vienna, Austria; theresa.bengough@goeg.at; 5School of Medicine and Population Health, University of Sheffield, Sheffield S1 4DA, UK; shoba.dawson@sheffield.ac.uk; 6Department of Nursing and Midwifery, University of Antwerp, 2000 Antwerpen, Belgium; kristel.paque@uantwerpen.be; 7Centre of Expertise—Sustainable Business and Digital Innovation, Campus De Ham, Thomas More University of Applied Sciences, 2800 Mechelen, Belgium; sarah.talboom@thomasmore.be; 8Learning Centre Désiré Collen, KU Leuven Libraries—Location: 2Bergen, KU Leuven, 3000 Leuven, Belgium; krizia.tuand@kuleuven.be (K.T.); thomas.vandendriessche@kuleuven.be (T.V.); 9Centre of Expertise—Care and Well-Being, Campus Zandpoortvest, Thomas More University of Applied Sciences, 2800 Mechelen, Belgium

**Keywords:** inclusive communication, crisis communication, pandemics, rapid review

## Abstract

**Background:** Crisis communication might not reach non-native speakers or persons with low literacy levels, a low socio-economic status, and/or an auditory or visual impairments as easily as it would reach other citizens. The aim of this rapid review was to synthesize the evidence on strategies used to improve inclusive pandemic-related crisis communication in terms of form, channel, and outreach. **Methods:** After a comprehensive search and a rigorous screening and quality assessment exercise, twelve comparative studies were selected for inclusion in this review. Data were analyzed and represented by means of a structured reporting of available effects using narrative tables. **Results:** The findings indicate that a higher message frequency (on any channel) may lead to a lower recall rate, audio–visual productions and tailored messages prove to be valuable under certain conditions, and primary healthcare practitioners appear to be the most trusted source of information for most groups of citizens. Trust levels were higher for citizens who were notified in advance of potential exceptions to the rule in the effect of preventive and curative measures promoted. **Conclusions:** This review contributes to combatting information inequality by providing evidence on how to remove the sensorial, linguistic, cultural, and textual barriers experienced by minorities and other underserved target audiences in COVID-19-related governmental crisis communication in response to the societal, health-related costs of ineffective communication outreach.

## 1. Introduction

Access to information is not only a universal human right [1], but also key in combating a crises, such as the COVID-19 pandemic [2]. Yet, various signals indicate that current crisis communication does not reach all target groups equally. This was the case during the recent COVID-19 pandemic. Minority groups and people who experience sensorial, linguistic, cultural, or textual barriers, in particular, were at risk of not accessing government communication [3]. Consequently, crisis communication might not reach non-native speakers or persons with low (health) literacy levels a low socio-economic status (SES), and/or an auditory or visual impairments as easily as it would reach other citizens. Indirectly, inefficient communication outreach increases the social- and health-related costs of the care system, e.g., through the late notification of an emergency situation or because of the mistrust of citizens regarding governmental strategies to solve healthcare problems effectively. This was particularly the case in low-literacy populations during the COVID-19 pandemic.

An inclusive crisis communication approach is successful when it does reach citizens of all abilities. This can be achieved by prioritizing the following four aspects of a crisis communication policy [2,4]: (1) accessible forms of communication (including (re)translations and media access services, such as subtitling, sign language, easy read, symbols, etc.), (2) accessible channels of communication (online or offline), (3) an efficient spread, and (4) effective outreach.

The aim of this rapid review was to synthesize evidence on the strategies used to improve inclusive pandemic-related crisis communication in terms of form, channel, and outreach. The project was initiated during the COVID-19 crisis. Ahl et al. [5] define crisis or risk communication as “the open two-way exchange of information and opinion about risk, leading to better understanding and better (clinical) decisions.” We focused particularly on strategies that acknowledge the (multi)linguistic and sociocultural diversity, sensory limitations, and degree of literacy of the world population. We operationalized the term “inclusive” as the removal and overcoming of sensorial, linguistic, cultural, and textual barriers to access and absorb information in the wider context of the crisis communication process focusing on pandemics.

### 1.1. Pandemic Context

Crisis communication is event-specific and can happen before, during, or after an unexpected or unanticipated event or disaster [6]. With respect to natural disasters, a distinction can be made between biological natural disasters (e.g., epidemics and pandemics) and weather-related natural disasters (e.g., volcanic eruptions, floods, tsunamis, droughts, tornados, earthquakes, wildfires, landslides, etc.). They typically differ in scale, duration, and intensity. Weather-related natural disasters are generally limited in time and demand attention and action over a relatively short period of time [7]. Biological natural disasters, on the other hand, are long-lasting and require attention and action over much longer periods of time [8]. These differences also influence the emotional response and sensitivity of the population with respect to both types of natural disasters [6,9]. This project studies inclusive communication in the context of biological natural disasters, more specifically in a pandemic or epidemic context. Pandemics are characterized by the widespread outbreak of infectious diseases, which can spread rapidly across populations and geographical regions. This spread is influenced by various environmental factors, such as climate change, deforestation, urbanization, and wildlife–human interactions [10]. These factors can alter the habitats of pathogens and vectors, facilitating the transmission of diseases to humans [11]. Public health systems must therefore be prepared to detect, monitor, and respond to such outbreaks to prevent widespread morbidity and mortality. Other contexts requiring crisis communication, such as natural disasters, but also environmental or ecological disasters (e.g., oil spills, chemical waste dumps, the dioxin crisis, the Chernobyl 1986 and Fukushima 2011 nuclear meltdowns, etc.) and terrorism, were not taken into account for this rapid review.

### 1.2. Crisis Communication

Form, channel, and outreach are important preconditions for an inclusive crisis communication process. Such processes encompass additional elements that, while not the focus of this project, need to be taken into account as part of the crisis communication context. A first element is message content: what information is included in the message and how is it framed? A second element is behavior change. Some crisis communication messages are informative in nature; other messages aim to actively achieve behavior change—for instance, to convince people to take vaccines or to follow guidelines on safety measures. In this review, we focus on the short-term goals of accessibility and reach and how they is influenced by the roles of form, channel, and outreach potential in conveying a message.

Form refers to the different modalities the message content takes, i.e., written text, video, infographics, subtitling, translations, audio, and audio description (an additional narrative voice that provides information about relevant visual elements in a media work for people with visual impairments). Translation includes traditional interlingual translation from one language and culture into another, intralingual translation within the same language, and forms of intersemiotic translations (from one modality, e.g., written words, to another, e.g., spoken language or visuals).

Channel refers to the medium used to distribute the message and its different forms, including online and print channels. For example, printed folders, posters, television, radio, fixed phones, mobile phones, text messages and SMS, as well as internet-based resources, such as email, video conferencing, social media (WhatsApp, Facebook, Twitter, etc.), and (government) websites.

Outreach refers to the appropriateness of a message’s form and channel for distribution to the target audience. This is a precondition to achieve wider access to information for the intended target groups, which indirectly supports a larger outreach and exposure in the long term.

### 1.3. Prioritized Target Groups

To achieve an inclusive approach in crisis communication, we paid attention to specific groups that are at risk of experiencing persisting barriers to access information and/or because they have a low socio-economic or literacy status. Different types of barriers considered in this review are:

Sensorial barriers: barriers to access the message content due to a permanent or temporary visual or auditory impairment, such as blindness, hearing loss, or deafness. For example, a hard-of-hearing person cannot access press conference videos without subtitling.

Linguistic barriers: barriers to access the message content due to linguistic accessibility problems. This includes low literacy skills and the level of language proficiency. For example, someone speaking a foreign language or not mastering the official language at a proficient level (i.e., language learners) may not understand a text’s original language, no matter in which domain of society it is situated.

Cultural barriers: barriers to access the message content due to a different cultural background (e.g., different values and belief systems, behavioral patterns, and communication practices). Non-verbal textual aspects (such as the use of colors, images, symbols, reading order, body language, gestures, etc.) may differ from one cultural linguistic community to the next, and may influence how a message is accessed or received. For example, not identifying with the role models used in a video due to generational or cross-regional differences.

Textual barriers: barriers to access the message content due to the complexity and/or lack of clarity of the message. Both can constitute a potential barrier, particularly when a source text is used as a basis for translation for different audiences (=target text). If the source text is not clear, easy to understand, and quick to process without an extensive cognitive effort of the receiver, the message will not come across efficiently. In addition, if the source text is overly complex and/or unclear, its potential (re)translations will also remain unclear. 

## 2. Why Is This Review Important?

In the event of a biological natural disaster, such as a pandemic or an epidemic, citizens actively seek information on how to act and deal with the imminent threat. What is more, as long as no treatments or vaccines are available, the control of a pandemic relies entirely on public health interventions, such as social distancing, contact tracing, mask wearing, and lockdowns [12]. Public access to information—the availability and accessibility of timely, high-quality information—is therefore vital for combating the outbreak of an infectious disease and “flattening the curve”. According to Koinig [13], the government plays a crucial role in managing a pandemic crisis by raising public awareness of the health threat and providing the population with targeted and timely information about the various containment and mitigation measures that are being imposed. Good crisis communication informs, instructs, and motivates; it builds trust for the authorities [14] and dispels rumors and misinformation. In addition, it empowers citizens in the sense that they know what to do to avoid and deal with infection. This requires intensive communicative efforts and effective communication strategies. Most importantly, these efforts and strategies should meet the specific communication needs of all populations to ensure that all societal groups are able to access, understand, and comprehend the information being communicated [15,16]. Indeed, as Hyland-Wood [17] observe, “there is no ‘one size fits all’ communications strategy to deliver information during a prolonged crisis”. To fulfill the aims of inclusive crisis communication, all groups of citizens should be included and involved. The crisis communication should be targeted, designed, and adapted to their various needs. Recent studies on the topic of crisis communication during the COVID-19 pandemic have shown, however, that not all groups of citizens are reached equally [18,19]. There has been a disproportionate toll on vulnerable populations as most governments have failed to customize their crisis communication to these particular target groups. Some citizens have special needs and thus experience difficulties in accessing correct information, leading to an asymmetry of information where they might be less informed than others. This, in turn, can result in unequal disease prevention protection across society [16]. In what follows, we spell out what is already known from a variety of different study types on how to potentially remove sensorial, linguistic, cultural, and textual barriers. This rapid review further reports on the results from comparative study designs.

### 2.1. Removing Sensorial Barriers

One group that is particularly vulnerable in crisis times are people with disabilities. Previous studies have shown that it is much harder to reach out to and communicate with these people in crisis times [20,21]. This is only complicated by a general lack of knowledge amongst governments, authorities, municipalities, and companies about how to meet the needs of disabled people. As such, people with disabilities run a much higher risk of being disproportionately affected by a crisis [22]. For this rapid review, we focused on people with a permanent or temporary visual or auditory impairment, such as blindness, hearing loss, or deafness. We did not focus on people with a mental illness, neuropsychiatric disability, or mobility impairment, although similar disadvantages may be present in these groups as well. The American Association on Health and Disability (AAHD) conducted an online survey on COVID-19 and disability during the first wave of the pandemic [23]. The survey included a set of questions on the preferred channels for accessing information about the COVID-19 pandemic. Of the deaf and hard-of-hearing people, “34% [...] said the Internet was the most important source of information, followed by Television (26%) and Health Care Providers and Relatives (21%)”. For persons with a visual impairment, “33% of respondents said the Television was the most important source of information, followed by the Internet (28%) and Radio (15%)”. This is in line with another survey performed by Holloway [24] who observed that blind and visually impaired people in Australia accessed information about the COVID-19 pandemic mostly through television and radio news (with government and health institutions being the most popular sources of information). Naylor [25] studied the effects of the COVID-19 lockdown in Glasgow on people with hearing loss. They indicated that hard-of-hearing people experienced difficulties conversing with people wearing face masks due to muffled sounds and a lack of speech-reading cues. Naylor et al. [25] therefore suggested the adoption of transparent face masks to alleviate some of the communicative difficulties experienced by this population. The same suggestion was made by Mörchen, Kapoor, and Varughese [26] in a study on communication with visually impaired people and eye health patients during the pandemic. Although people with hearing loss did not experience major obstacles when following TV and radio updates about the evolving pandemic, Naylor et al. [25] nevertheless suggested the use of live subtitles on video calls. This suggests that much can be learned from studying the tactics and strategies proposed by the target group for conveying messages in a pandemic context.

### 2.2. Removing Linguistic Barriers

Another group that faces considerable challenges in crisis times are foreign language speakers. In our super-diverse societies, foreign language speakers and language learners may not always master the local official language(s) at a sufficiently high proficiency level to understand the government’s crisis communication messages. Multilingual crisis communication (i.e., the translation of crisis communication messages into various languages) is therefore an important prerequisite to bridge these language barriers and to ensure that the entire population of a country is reached. Although the role and importance of language translation and multilingual crisis communication in multilingual and multicultural societies has been highlighted before [27], it remained underestimated, if not unrecognized during the COVID-19 pandemic. In a recent study, which aimed at assessing the inclusion of individuals with a migrant background in COVID-19 prevention measures, Maldonado [28] investigated whether governmental risk communications were available in common migrant languages across Europe. They identified clear gaps in the availability of translated COVID-19 risk communications across Europe, excluding migrants from receiving the necessary information in their own languages. Chen [29] explored the availability of multilingual public health messages against the spread of COVID-19 in Taiwan between January and April 2020, with similar results. Also, indigenous populations faced significant language barriers, and were thus excluded from most public health communications. The identified reasons for this include the dominance of English-centric global mass communication, the longstanding devaluation of minoritized languages, and the failure to consider the importance of multilingual repertoires for building trust and resilient communities [30].

### 2.3. Removing Cultural Barriers

Different strategies have been developed for optimal crisis communication in a culturally diverse society [31]. This could involve the use of different channels and communication platforms, differences in the speed of speaking, eye contact with the audience, facial expressions, and differences in tone of voice (e.g., an empathetic, compassionate, or supportive tone versus a serious, clinical, or reserved tone). Wertz and Kim [32], for example, observed that the Korean government uses a more aggressive message strategy than the US government in times of crisis. Similar differences were observed by Low [33] between the communication strategies of Western and Asian governments. According to Oliveira [34], culturally adjusted crisis strategies are not yet sufficiently adopted. Failure to consider cultural factors may lead to offensive feelings, misunderstandings, criticism, and an unwillingness to follow the various mitigation and containment measures.

### 2.4. Removing Textual Barriers

A good strategy to remove textual barriers is the use of plain language or easy-to-read language. Plain language (also called plain writing or plain English) is a style of writing that is easier to read, understand, and use, compared to normal language, as it avoids verbose, convoluted language and jargon. It is used to reach all audiences. Easy-to-read language on the other hand is specifically designed to meet the needs of people with cognitive and learning disabilities, as well as language learners or people with low literacy levels. But also migrants, people with severe social problems, or the elderly can benefit from easy-to-read language [35]. The World Health Organization (WHO) observes that, if people have to read a “message several times to understand it, they are not likely to act on the advice and guidance in the message” [36]. To that aim, the WHO has suggested (1) to organize information so the most important points come first, (2) to create a single overarching communication outcome (SOCO) that defines the desired outcome, for example, behavior change, (3) to break long and complicated information into understandable portions, (4) to use simple language to explain the meaning of technical terms, and (5) to format documents with plenty of white space so they are easy to read. Although it has become standard practice to translate crisis communication messages into plain or easy-to-read language, very little research has been performed on the readability of COVID-19 crisis communication messages. One exception is Basch [37] who assessed the readability of information posted on the Internet about the COVID-19 pandemic. Multiple readability tests were conducted on 100 different English language websites, including the Coleman–Liau Index (CLI), the Gunning Fog Index (GFI), the Simple Measure of Gobbledygook (SMOG) Grade Level, the Flesch–Kincaid Grade Level (FKGL), and the Flesch–Kincaid Reading Ease (FRE). To have a maximum impact, crisis communication messages should be readable at the 6th-grade reading level [38]. Four of the five measures (CLI, GFI, SMOG, and FRE) found that readability on these websites exceeded the 10th-grade reading level, indicating that these texts would be difficult to read for the average American.

## 3. Objectives and Review Question

To reach all groups equally, inclusive crisis communication strategies are needed, which focus on removing or responding to various sensorial, linguistic, cultural, and/or textual barriers. The Emergency Risk Communication Model by Seeger [39] highlights optimal accessibility and exposure as necessary conditions to achieve the longer-term goal of behavior change (i.e., the willingness to be tested and/or vaccinated, or to follow the various containment and mitigation measures that are imposed). In this rapid review, we focus on the short-term goals. The review question is as follows:

For persons with sensorial, linguistic, cultural, and/or textual barriers, which communicate interventions on the level of the form, channel, and outreach in crisis communication messages are the most effective and applicable in an epidemic or pandemic context, from a comparative perspective?

## 4. Methods

### 4.1. Design

For this rapid review, we followed the principles and guidelines in the WHO Practical Guide on Rapid Reviews to Strengthen Health Policy and Systems [40]. The rapid review protocol consists of several steps in two different review phases. In the first phase, relevant papers published until 17 May 2021 were retrieved from 7 major electronic databases, screened, and assessed for quality. The first phase resulted in 9796 retrieved studies. After the removal of duplicates, systematic reviews, and meta-analyses, 5825 studies were eligible for screening. However, since the first phase resulted in a very limited number of relevant papers, considering the increased popularity of crisis communication as a topic during the current global pandemic, a rapid review update was initiated during the project time in an attempt to find additional sources at a later stage (phase 2). In phase 2, we retrieved, screened, and assessed papers that were published between 17 May and 15 October 2021. We retrieved 2507 new studies published after 17 May 2021. After the removal of duplicates, systematic reviews, and meta-analyses, 1675 studies remained for screening. We followed the same methodological strategy for both phases (outlined in Figure 1).

### 4.2. Inclusion Criteria

Population of interest: Priority was given to evidence that was relevant to people in the following situations: (a) foreign language speakers and/or (b) citizens with low literacy skills and/or (c) citizens with a low SES and/or (d) citizens with an auditory or visual impairment.

Intervention of interest: Priority was given to communicative interventions on the level of the form, channel, and outreach capacity in crisis communication. Studies focusing on the content of the message were excluded.

Comparison: Priority was given to standard crisis communication, or other interventions in relation to form (e.g., subtitles versus voiceover and static versus dynamic pictorial language), channel (e.g., online versus print) or outreach, or no communication at all. Only comparative study designs were taken into account, e.g., evaluation studies, clinical, intervention, observational, comparative, before and after, and preventive studies, as well as RCTs, quasi-RCTs, and other types of controlled studies. Non-comparative and qualitative studies, dissertations, conference papers, books, editorials, and opinion pieces were excluded.

Outcome of interest: Priority was given to inclusive crisis communication markers, such as accessibility and exposure. Accessibility refers to the ideal situation in which all sensorial, cognitive, linguistic, and cultural barriers have been overcome. Exposure refers to the situation in which the actual outreach potential of the communication strategy has been achieved in the target group. Studies focusing on behavior change as an outcome were excluded (as this is a long-term goal).

Context of interest: This rapid review was written in the context of pandemic or epidemic crisis situations. The World Health Organization [41] has identified numerous infectious diseases that have the potential to become international threats. Based on their work, we compiled an initial list of 23 pandemic or epidemic diseases. We subsequently removed all zoonotic diseases. No distinctions were made on the basis of the pathogen (e.g., virus or bacterium) or whether the disease is spread via saliva or aerosols. This resulted in a final list of 12 pandemic or epidemic disease contexts we focused on: Ebola Virus Disease, Lassa Fever, Avian Flu, Influenza, Seasonal Influenza, Pandemic Influenza, Middle East Respiratory Syndrome (MERS), Meningococcal Meningitis, Hendra Virus Infection, Nipah Virus Infection, Novel Coronavirus (2019-nCoV), Severe Acute Respiratory Syndrome (SARS), and Smallpox.

### 4.3. Search Strategy and Study Retrieval

We searched 7 major electronic bibliographic databases for relevant papers: CINAHL (EBSCO), Web of Science Core Collection (including the ISI Social Science Index and Arts and Humanities Index), Medline/PubMed, Embase, ERIC (OVID), Cochrane CENTRAL, and Cochrane CDSR. The search strategy was developed in collaboration with three biomedical reference librarians from the KU Leuven Libraries—Location: 2Bergen—learning Centre Désiré Collen (Leuven, Belgium). The full search strategy and complete list of search terms applied to Medline/PubMed can be found in the annex (see Appendix A). We used four parameters to build our search strategy. The first search string thus consisted of all terms that capture the epidemic and/or pandemic context. The second and third search strings included terms that characterize interventions in crisis communication (Phenomenon) and the topical areas Channel and Form. The fourth search string represented the study designs under review. The search strings were adapted for use in the other databases. For each of the four parameters, we looked for specific Medical Subject Heading (MeSH) terms, synonyms, and related terms. This rapid review included two phases of study identification, which allowed us to include additional studies produced during or in response to the ongoing COVID-19 crisis situation. The search results from included databases were exported and merged into the citation management software EndNote (version X9), yielding a total of 9796 retrieved studies for phase 1 (up to 17 May 2021) and 2507 studies for phase 2 (from 18 May to 15 October 2021). We sorted the retrieved findings. After the removal of 3049 duplicates, an additional 1754 review projects and meta-analyses were removed. This left us with 7500 unique studies across both phases.

We only considered studies written in the English language, mainly because new evidence illustrates that including additional, non-English literature does not seem to change the conclusions to a large extent [42]. Nussbaumer-Streit, in her methodological review, states that non-English publications are not always the main publication (and usually of a smaller scale) and/or do not seem to alter the size or direction of an effect measured to a large extent. Their exclusion was therefore promoted as a viable methodological shortcut in the context of rapid reviews. Studies had to be full-length articles or papers that were peer-reviewed. Studies were initially considered fitting when they focused on foreign language speakers and/or people with low literacy skills, and/or people with low SES and/or people with an auditory or visual impairment. During the first scoping and screening phase of the included papers, we noticed that only a low number of comparative studies focused on the included target groups. We therefore revisited the inclusion criterion "population of interest". Studies that did not focus on these specific target groups but met all other criteria were picked up and considered as indirect evidence to inform practice and policy. It was decided to highlight potential extrapolation issues from one population to another as part of the findings and Discussion Section.

We followed the best practices guidelines for abstract screening, as outlined by Polanin et al. [43], and used Rayyan, a web and mobile screening app for systematic reviews [44], to facilitate screening. In order to avoid random and/or systematic errors in the study selection, and in order to ensure that the above eligibility criteria were applied consistently, a double-screening approach was adopted for a subset of the studies. We opted for an approach in which 20% of all papers was double-screened and interrater agreement rates were calculated. The interrater agreement (also known as percent agreement) is defined as the degree to which scores/ratings between reviewers looking at the same abstract are identical [45]. In judging the outcome of the interrater agreement, we made a distinction between major and minor conflicts. A major conflict occurs when screener A has included a study, whereas screener B has excluded the study. A minor conflict occurs when screener A has either included or excluded a study, whereas screener B remains undecided (by answering “maybe”). In our case, individual agreement rates varied between 88% and 100% for phase 1 (first batch of studies retrieved), with an average group agreement of 96%. This bolstered our confidence in the individual screening results. In addition, the majority of conflicts was minor conflicts (67% to be precise, and up to 88% if the conflicts with screener 8 out of 14 screeners were ignored). The deviant results of screener 8 were related to the incapability of assessing the study design criterion. A total of 17 major and 32 minor conflicts was observed. In phase 2, we reached an agreement rate of 95%, with only minor conflicts. In both cases, discrepancies were resolved by a third, experienced reviewer.

### 4.4. Quality Appraisal

Four independent reviewers screened the full texts of the 26 (phase 1) and 22 (phase 2) remaining studies to further identify the eligibility of the articles (see Appendix B, Table A1 and Table A2).

The quality of the studies was simultaneously assessed. Given the broad variety in comparative designs encountered, only two quality aspects were considered: (1) Is the basic study design valid for comparative purposes? And (2) is the study methodologically sound? (in the sense of being executed according to the state of the art for a particular design). Disagreements between two reviewers were resolved by discussion and consensus. A total of 21 studies (phase 1) and 15 studies (phase 2) eventually did not meet the eligibility criteria when considering the full text. Most were excluded due to wrong study design (non-comparative/non-evaluative), although many also lacked a correct topic, focus, context, or population. In the end, only 12 studies (5 for phase 1 and 7 for phase 2) were deemed eligible. All of these met the two listed basic parameters of quality.

### 4.5. Analytical Approach

Because of the topical and methodological diversity displayed in the studies, we opted for a structured reporting of available effects using narrative tables [46]. For each eligible study, a summary sheet was made. The summary sheet included information on the author, year and country, study design, sender and receiver (i.e., population) of the crisis communication messages, intervention, control, and outcome of the scientific evidence, and were deemed necessary, concluding remarks. A final column with remarks from the reviewers provided specific information to readers that might help facilitate their assessment of the relevance of the study for their particular local context and target group. Results were aggregated in a narrative storyline on the effectiveness of form, channel, and sender.

## 5. Results

A total of 12 studies was included in this review (Box 1).

We briefly summarize the findings of the 12 included studies with regard to the effectiveness and applicability of different crisis communication strategies that target the inclusion of these vulnerable and minority groups (see also the summary sheets in Appendix C, Table A3, Table A4, Table A5, Table A6, Table A7, Table A8, Table A9, Table A10, Table A11, Table A12, Table A13, Table A14 and Table A15). We evaluated the impact of form, channel, and sender in crisis communication on three different outcomes: spread, reach, and awareness levels of citizens. More specifically, we focus on those citizens experiencing sensorial, linguistic, cultural, and/or textual barriers.

### 5.1. Effectiveness of Form

In terms of form specifications, several authors suggest that video messages increase knowledge about crisis situations and the measures to be taken [47,48,49]. However, certain forms have a greater effect than others. Bekalu et al. [50] indicate that non-narrative, didactic messages convey information in a health crisis better than messages in a narrative form. However, the study findings need to be read with caution as the direction of the effect may have been influenced by factors such as the choice of the narrative clip included in the study (e.g., a film clip that was not well understood). Mistree et al. [47] argue that videos with concept explanations significantly increase pandemic knowledge compared to videos that only provide facts. This is in line with the findings from Lee’s and Jahng’s [51] study in which the effect of storytelling on levels of trust, perception of crisis responsibility, and persuasion has been judged as positive. Longer videos of approximately 20 min score better in terms of increasing knowledge compared to shorter ones. A side note to this finding is that this effect largely depends on the length of the average attention span of the public or context for which the video is made [47]. We also found evidence for the effectiveness of wordless, animated videos [49]. This is promising for an outreach to people with linguistic or textual barriers. An important positive side effect of using infographics displaying a particular image of scientists is that depicting them as a normal person slightly increases the believability in the narrative brought [52]. The result was not significant though.

Box 1Final list of included studies in alphabetical order [47,48,49,50,52,53,54,55,56,57,58,59].*Agley, J., Xiao, Y., Thompson, E. E., & Golzarri-Arroyo, L. (2021). Using infographics to improve trust in science: a randomized pilot test. BMC research notes, 14(1), 1–6.*Bahety, G., Bauhoff, S., Patel, D., & Potter, J. (2021). Texts Don’t Nudge: An Adaptive Trial to Prevent the Spread of COVID-19 in India.*Baseman, J. G., Revere, D., Painter, I., Toyoji, M., Thiede, H., & Duchin, J. (2013). Public health communications and alert fatigue. BMC health services research, 13, 295.*Baseman J., Revere D., Painter I., Oberle M., Duchin J., Thiede H., Nett R., *MacEachern D., Stergachis A. (2015) A Randomized Controlled Trial of the Effectiveness of Traditional and Mobile Public Health Communications With Health Care Providers. Disaster Med Public Health Prep. Feb, 10(1), 98–107.*Bekalu M.A., Bigman C.A., McCloud R.F., Lin L.K., Viswanath K. (2018). The relative persuasiveness of narrative versus non-narrative health messages in public health emergency communication: Evidence from a field experiment. Preventive Medicine 111, 284–90. https://doi.org/10.1016/j.ypmed.2017.11.014*Chen, L., Tang, H., Liao, S., & Hu, Y. (2021). e-Health Campaigns for Promoting Influenza Vaccination: Examining Effectiveness of Fear Appeal Messages from Different Sources. Telemedicine and e-Health.*Dennis, A. S., Moravec, P. L., Kim, A., & Dennis, A. R. (2021). Assessment of the Effectiveness of Identity-Based Public Health Announcements in Increasing the Likelihood of Complying With COVID-19 Guidelines: Randomized Controlled Cross-sectional Web-Based Study. JMIR public health and surveillance, 7(4), e25762.*Johnson, B. B., & Slovic, P. (2015). Fearing or fearsome Ebola communication? Keeping the public in the dark about possible post-21-day symptoms and infectiousness could backfire. Health, Risk and Society, 17(5), 458–471.*Mistree, D., Loyalka, P., Fairlie, R., Bhuradia, A., Angrish, M., Lin, J., … & Bayat, V. (2021). Instructional interventions for improving COVID-19 knowledge, attitudes, behaviors: Evidence from a large-scale RCT in India. Social Science & Medicine, 276, 113846.*Okuhara T., Okada H., Kiuchi T. (2020). Examining persuasive message type to encourage staying at home during the COVID-19 pandemic and social lockdown: A randomized controlled study in Japan. Patient Education and Counseling 103(12), 2588–93. https://dx.doi.org/10.1016%2Fj.pec.2020.08.016*Torres, C., Ogbu-Nwobodo, L., Alsan, M., Stanford, F. C., Banerjee, A., Breza, E., … & COVID-19 Working Group. (2021). Effect of physician-delivered COVID-19 public health messages and messages acknowledging racial inequity on Black and White adults’ knowledge, beliefs, and practices related to COVID-19: a randomized clinical trial. JAMA Network Open, 4(7), e2117115-e2117115.*Vandormael, A., Adam, M., Greuel, M., Gates, J., Favaretti, C., Hachaturyan, V., & Bärnighausen, T. (2021). The effect of a wordless, animated, social media video intervention on COVID-19 prevention: online randomized controlled trial. JMIR public health and surveillance, 7(7), e29060

In addition, evaluating the impact of open and proactive communication in crisis situations was within reach of this review. We retrieved a study from Johnson et al. [53] emphasizing the importance of communicating in advance, the reason behind a particular measure when it is implemented. In case of an exceptional situation (such as symptoms of disease occurring after a quarantine period), it is advisable to mention it up front and to communicate why a certain measure (in this case, the length of a quarantine period) was chosen. Informing people in advance about possible exceptional situations that might occur increased the trust levels of citizens in health experts and institutes. Explaining the reason behind measures taken in response to these exceptions also had a positive effect on the trust levels of citizens. Trust levels were lower in the comparative group that spontaneously encountered these exceptions.

### 5.2. Effectiveness of Channel

In terms of channel specifications (i.e., which medium is used to disseminate information), Baseman et al. [54] suggest that messages sent by email generate higher recall rates than messages sent by other means, such as fax or SMS. The scientific evidence, however, is not entirely unequivocal. If one only considers the situations where the messages are actually received (as compared to the situations where messages are sent out but fail to reach the receiver), the recall rates for email and fax are identical (48.3%). Overall, the lowest recall occurs for messages sent via SMS.

In terms of the effectiveness of message frequency and timing, the evidence suggests that a higher message frequency (on any channel) may lead to a lower recall rate [55]. Bahety et al. [56] further suggest that timing makes a difference: when messages are sent too late (long after an outbreak) and without much visual support, they might lose their effect.

### 5.3. Effectiveness of Sender

We also looked at the effect of using different senders to disseminate relevant health information in a crisis situation, with a particular focus on pandemics and epidemics. Evidence suggests that medical doctors are best placed to deliver such information, as they increase the willingness of citizens and patients to adhere to advice [57]. In some cases, though, other senders appear to be more effective. For example, respondents who received an emergency message from a COVID-19 patient or a resident of a COVID-19 outbreak area felt more vulnerable to the virus than respondents who received an emergency message from a doctor. Overall, citizens seem to trust information from senders whose identity or institute can be verified better [58]. It also positively influences people’s search intent. Messages tailored to religious, economic, or other specific identities also increase the motivation to follow measures [59]. Yet, investigations studying the impact of skin color or differences in the ethnic profiling of senders, such as doctors, do not suggest any beneficial effect on knowledge increase [48].

## 6. Discussion

In this review, we described the findings of existing research on inclusive crisis communication, informed by the literature from cognitive and social psychology, sociology, health sciences, and applied forms of crisis communication. This rapid review was set up from an inclusivity perspective. People’s engagement with and response to public health information and messaging appears to be influenced by their cultural and social identity, age, gender, and access to resources [17]. In addition, there is great individual variation in the needs of people living in challenging circumstances. We focused on crisis communication for a specific proportion of underserved target groups in pandemic or epidemic conditions. Specific attention was paid to the needs of people living in challenging circumstances and/or minority groups who experienced sensorial, linguistic, cultural, or textual barriers, i.e., non-native speakers, people with low literacy skills, people with low SES, and people with sensory (auditory or visual) impairments.

The COVID-19 pandemic has been an important catalyst for recent scholarly work on how to render crisis communication accessible to all, outlining various strategies, policies, and recommendations, tailored to diverse audiences. However, both the state of the art and the results from this rapid review testify to a general lack of high-quality academic research on the topic of inclusive crisis communication. Indeed, of the 7500 retrieved original studies, only 12 studies made it through the entire screening process and quality appraisal. This is indicative of an important gap in the literature for high-quality, comparative study designs on the topic of inclusive crisis communication that needs to be tackled in future research. This gap may also help explain why (inter)nationally validated policy guidelines on inclusive crisis communication hardly exist to support governments in reaching the goals of inclusivity. The fact that many researchers were conducting experimental and comparative studies in the midst of the COVID-19 pandemic, with the request to deliver fast results, may have contributed to the low-quality level detected in the full study pool and the overall lack of robust evidence from high-quality randomized controlled studies.

Another reason for the initially very-small sample of included studies was our specific focus on a population of non-native speakers, people with low literacy skills, people with low SES, and people with auditory or visual impairments. Many potentially relevant studies targeting citizens more generally were initially excluded on the basis of a wrong population, but picked up again in a later phase of the review to increase the number of studies that can provide relevant information to end-users, for example guideline developers. Using a selective sampling procedure, we reselected studies that were excluded for population purposes (read: studies that did not focus on the vulnerable populations as specified in our inclusion criteria). Despite the fact that the conclusions from these studies only provide indirect evidence, it allows interested end-users to investigate the possibility of extrapolating insights from one population to another in their particular decision-making context. Working on review projects in crisis situations where evidence is generated based on progressive insight requires a substantial amount of flexibility and where an adaptation of predefined criteria is needed that might only work in ideal circumstances, but not necessarily for the context for which rapid systematic reviews need to be developed.

Given the importance of communication in a public healthcare context and the high economic and social costs of ineffective communication strategies, Dreisbach and Mendoza-Dreisbach [60] plead for a new field in linguistics to tackle public health translation in emergencies: emergency linguistics. In response to the current lack of multilingual crisis communication, several recommendations have been proposed. Maldonado et al. [28] recommend working with non-governmental organizations (NGOs) and migrant community groups. O’Brien et al. [61] concur that one should “establish strategic partnerships with relevant not-for-profit organisations in advance of crises so that communities are more likely to receive crucial information more rapidly and that they might have a higher level of trust in that information”.

## 7. Conclusions

The findings of this rapid review are meant to support researchers that take the (multi)linguistic and sociocultural diversity or degree of literacy of citizens into account in their work. Specifically, this review contributes to combatting information inequality by providing evidence on how to remove the sensorial, linguistic, cultural, and textual barriers experienced by minorities and other harder-to-reach target audiences in COVID-19 related governmental crisis communication. Our findings highlight the importance of form, channel, and sender in crisis communications in order to be inclusive and to reach underserved audiences. Although the content of a message may remain stable, content creators should reflect on which type of person (sender) is perceived as the most trustworthy by people in particular minority groups. They should also pay attention to the clarity, length, and accessibility of the content (form). The latter might require more contextual information regarding messaging for those with hearing or visual impairments, as their exposure to different types of information providing such content might be more limited. Frequency of electronic messaging (channel) should be modest, and the timing should be right to reach the intended audience. To conclude, cultural sensitivity generally is appreciated by many of the populations under study in this review [16]. These populations can be perceived as a critical case to support decisions on the best possible strategy for crisis communication: when it works for those who encounter (multiple) barriers, it will surely work for a more general population of citizens.

## Figures and Tables

**Figure 1 ijerph-21-01216-f001:**
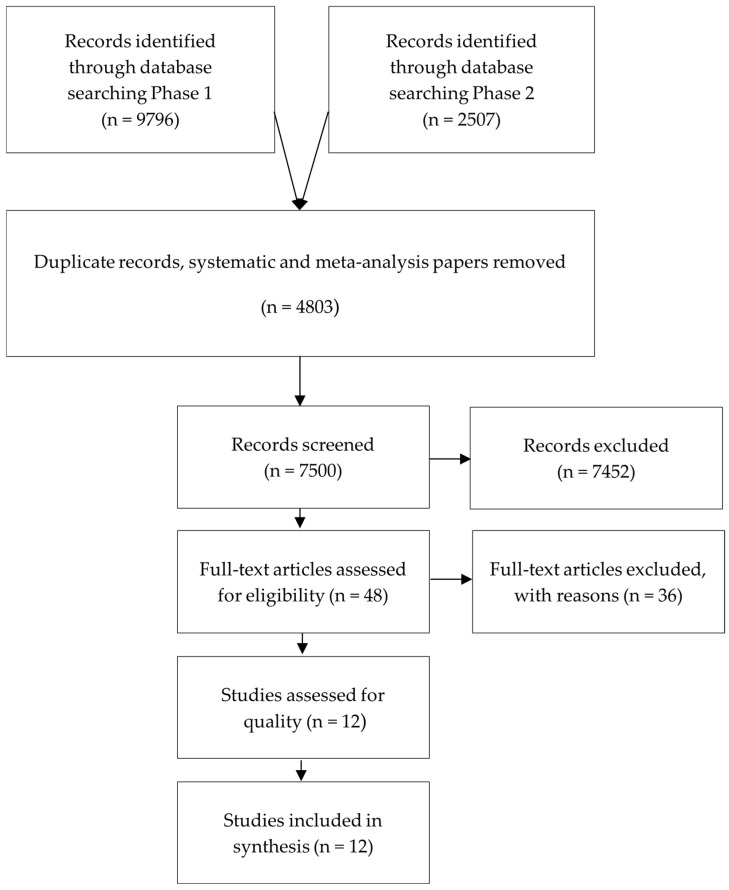
Outline of included and excluded studies.

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
