# Peer review of "Inclusive Crisis Communication in a Pandemic Context: A Rapid Review"

_ijerph, 2024, doi:10.3390/ijerph21091216_

Round 1

Reviewer 1 Report

Comments and Suggestions for Authors

The paper was excellent in every aspect.  The topic is important and needs studying, even further.  The article is very well structured and written.  Empirical research seems to be well done, well theoretically founded, and the results base themselves on the research, and are well documented.

This sentence is unclear, what is this "Bergen"?  The search strategy was developed in collaboration with three bio-375 medical reference librarians of the KU Leuven Libraries – 2Bergen – learning Centre Dé-376 siré Collen (Leuven, Belgium).

Result chapter has the number 3,should be 5

In discussion about Chen & al paper, one would like to know what meand verified and non-verified.

Some small issues I would anyway like to see corrected:

Many sections have homeless text.  I would like to see  subchapters follow right away the chapters above.  So, the subtitle 1.1, for example, would folow title 1, without any text inbetween.

About linguistic barriers I would comment that even native speakers find themselves at very different levels.  You should also discuss the aspect that simple and easy communication is most likely very often good and beneficial to non-impaired population groups.

You use the term "super-diverse society". Explain it a little, and give the origin.  How do you know when a society is super-diverse, and if all societies are, what is then the meaning of the word.

Some are fine, but some authors seem to miss the concept of paragraph.  I cannot understand that a whole subschapter would be just one paragraph, as for example in subchapters 2.1 to 2.4, or 4.2 before figure.

Figure 1 was badly situated and not fully readable in the manuscript.

Different authors handle English differently xxxx-related or xxxx related.  I would like to see uniform language use in this.

The following sentence was hard to understand: "Trust levels were higher in citizens who were notified in advance of 58 potential exceptions to the rule in the effect of preventive and curative measures promoted." I propose rewording.

Author Response

Comment 1: Bergen - We put 'location: 2Bergen' now to indicate that it is actually a place or address.

Comment 2: We clarified the meaning of verified and non-verified in relation to Chen's paper by adding the following explanation to the table: verified (badge that confirms authenticity and trustworthiness of the source).

Comment 3: We did not fully understand why the reviewer would suggest to put a subtitle under the introduction parts of our lead sections. We checked multiple other articles published in the journal and noticed various practices. We have added one subtitle in the methods part on the level of designs but left the other sections untouched.

Comment 4: Linguistics - we have added the critical case argument suggested by the reviewer to the conclusion section (last sentence to conclude the paper with).

Comment 5: super-diversity - We decided that we didn't need this argument to make our point and removed this phrase.

Comment 6: paragraphs under subtitles: In the outline of the template we suggested new paragraphs using the 'hanging' option for line spacing. We leave it up to the type setter to decide on the potential inclusion of white spaces instead, as suggested by the reviewer. 

Comment 7: Figure 1 was adapted.

Comment 8: We checked on the use of hyphens and removed them where appropriate. 

Comment 9: The phrasing has been revisited as follows: "Informing people in advance about possible exceptional situations that might occur increased trust levels of citizens in health experts and institutes.  Explaining the reason behind measures taken in response to these exceptions also had a positive effect on trust levels of citizens."

Reviewer 2 Report

Comments and Suggestions for Authors

This manuscript, "Inclusive Crisis Communication in a Pandemic Context: A Rapid Review," is dedicated to combating information inequality by providing evidence on how to remove sensorial, linguistic, cultural, and textual barriers experienced by minorities and other harder-to-reach target audiences in Covid-19 related governmental crisis communication, in response to the societal and health-related costs of ineffective communication outreach. For this rapid review, the authors followed the principles and guidelines outlined in the WHO Practical Guide on Rapid Reviews to Strengthen Health Policy and Systems (WHO, 2017).

The significance of this work lies in the authors' synthesis of evidence on strategies used to improve inclusive pandemic-related crisis communication in terms of form, channel, and outreach.

Comments and Suggestions for Authors:

• Generally, the conclusions are consistent with the evidence and arguments presented.

• Please describe the tools used to analyze the materials.

• It is recommended to transfer Table 1: Outcome of the assessment of the quality of the relevant studies in phase 1 and Table 2: Outcome of the assessment of the quality of the relevant studies in phase 2 to the Appendix.

• The authors stated: "We only considered studies written in English language, mainly because new evidence illustrates that including additional, non-English literature does not seem to change the conclusions to a large extent." Please provide further elaboration on this decision.

• Please discuss the results presented in Table 3: Summary sheet for Agley et al. (2021) within the text of the article. Additionally, it seems appropriate to move this table to the Appendix. 

Please describe in detail how your study fits for aims and scope of International Journal of Environmental Research and Public Health (IJERPH)

Author Response

Comment 1: We didn't use any tools. Since we already referenced our approach (McKenzie et. al., 2019), we provided more information about the subsequent step after completing the tables to section 4.5. "Results were aggregated in a narrative storyline on the effectiveness of respectively form, channel and sender."

Comment 2: In line with the comment from the reviewer, the tables included in the article that were indicated as supportive were moved to the appendix. 

Comment 3: We added the following information on the linguistic exclusion criteria: "Nussbaumer-Streit, in her methodological review, states that non-English publications are not always the main publication (and usually of a smaller scale) and/or do not seem to alter the size or direction of an effect measured to a large extent.  Their exclusion is therefore promoted as a viable methodological shortcut in the context of rapid reviews. "

Comment 4: We explained the case from Agley in more detail: "An important positive side effect of using infographics displaying a particular image of scientists is that depicting them as a normal person slightly increases the believability in the narrative they bring (Agley et al., 2021). The result was not significant though." 

Comment 5: We added a phrase to link the article to the mission of the journal under subsection 1.1.: "Pandemics are characterized by the widespread outbreak of infectious diseases, which can spread rapidly across populations and geographical regions. This spread is influenced by various environmental factors such as climate change, deforestation, urbanization, and wildlife-human interactions (Tazerji et. al., 2022). These factors can alter the habitats of pathogens and vectors, facilitating the transmission of diseases to humans (Bhattacharya et. al., 2020). Public health systems must therefore be prepared to detect, monitor, and respond to such outbreaks to prevent widespread morbidity and mortality."

Round 2

Reviewer 2 Report

Comments and Suggestions for Authors

I am satisfied with the feedback from the authors of the article